# Research on Selected Wildlife Infections in the Circumpolar Arctic—A Bibliometric Review

**DOI:** 10.3390/ijerph191811260

**Published:** 2022-09-07

**Authors:** Anastasia Emelyanova, Audrey Savolainen, Antti Oksanen, Pentti Nieminen, Olga Loginova, Khaled Abass, Arja Rautio

**Affiliations:** 1Thule Institute, University of Oulu & University of the Arctic, P.O. Box 7300, FI-90014 Oulu, Finland; 2Arctic Health, Faculty of Medicine, University of Oulu, P.O. Box 5000, FI-90590 Oulu, Finland; 3Finnish Food Authority (FINPAR), Elektroniikkatie 3, FI-90590 Oulu, Finland; 4Medical Informatics and Data Analysis Research Group, University of Oulu, P.O. Box 5000, FI-90014 Oulu, Finland; 5Laboratory of Parasite Systematics and Evolution, Center for Parasitology, A.N. Severtsov Institute of Ecology and Evolution, Russian Academy of Sciences, Leninskii Prospect 33, 119071 Moscow, Russia

**Keywords:** Arctic, bibliometric review, infectious disease, One Health, wildlife health, zoonosis

## Abstract

One Health, a multidisciplinary approach to public health, which integrates human, animal, and environmental studies, is prudent for circumpolar Arctic health research. The objective of our bibliometric review was to identify and compare research in select infectious diseases in Arctic wildlife species with importance to human health indexed in English language databases (PubMed, Scopus) and the Russian database eLibrary.ru. Included articles (in English and Russian languages) needed to meet the following criteria: (1) data comes from the Arctic, (2) articles report original research or surveillance reports, (3) articles were published between 1990 and 2018, and (4) research relates to naturally occurring infections. Of the included articles (total n = 352), most were from Russia (n = 131, 37%), Norway (n = 58, 16%), Canada (n = 39, 11%), and Alaska (n = 39, 11%). Frequently reported infectious agents among selected mammals were *Trichinella* spp. (n = 39), *Brucella* spp. (n = 25), rabies virus (n = 11), *Echinococcus* spp. (n = 10), and *Francisella tularensis* (n = 9). There were 25 articles on anthrax in eLibrary.ru, while there were none in the other two databases. We identified future directions where opportunities for further research, collaboration, systematic reviews, or monitoring programs are possible and needed.

## 1. Introduction

One Health is an approach to policies and research in which multiple disciplines work together to promote public health [1]. It calls for an integrated study of humans, animals (both domestic and wild), and the environment (Figure 1). The One Health paradigm is vital to understanding the complex relationships between health and disease in the circumpolar Arctic [2,3,4,5,6,7].

Humans and wildlife in the Arctic share a unique bond. For most of history, the survival of Indigenous peoples in the Arctic has been completely dependent upon hunting and fishing, due to the scarcity of edible plants [8,9]. Nowadays, humans in the Arctic are also dependent on semidomestic and domestic animals, both native and introduced. In recent decades, a rapidly changing climate and society are straining the balance between humans, animals, and the environment in the Arctic [9,10].

Human, domestic animal, and wildlife health are all challenged by rapid environmental changes, such as spreading infectious and vector-borne diseases [11], mobilization, and toxic substance exposure [12,13]. Risks for humans are assumed to depend on environmental contamination, infection prevalence rates in animal populations, and contact patterns between other humans, the environment, and animals [14]. In recent studies on human infections in the Arctic, it was found that tick-borne diseases, tularemia, anthrax, and vibriosis are most likely to be impacted by climatic factors, and increased temperature and precipitation are predicted to have the greatest impact on those infections [10,11,15,16,17]. Nowadays, especially after the outbreak of coronavirus disease 2019 (COVID-19) and the ensuing pandemic, biosecurity and zoonotic diseases are a central focus, e.g., in the projects of the Arctic Council [18].

However, infections are a natural part of life in the Arctic, as elsewhere. Thousands of different viruses, bacteria, parasites, and fungi contribute to the unique Arctic ecosystem, forming an essential part of its biodiversity alongside humans, animals, and plants [10]. While the consequences of these infections can range from asymptomatic—with no adverse effects on health—to fatal, understanding wildlife infections will also lead to a better understanding of human health. Changes in vectors of wildlife infections are expected and are already reported to have taken place due to climate change and due to the introduction of non-native infectious agents into the Arctic from migrating humans and domestic animals [10,11].

In order to identify key focus areas where research is lacking as well as topics warranting circumpolar monitoring, it is essential to first assess what research has already been published. There are about 30 Russian journals included in PubMed. Most of them publish only abstracts in English, while the full texts are in Russian [19]. On the other hand, medical and veterinary research published in Russian journals is found in the database eLibrary.ru. Russia comprises almost two-thirds of the Arctic human population and half of the territory. Therefore, it is crucial to know what has been published about infectious diseases in wildlife in the Russian Arctic.

The objective of this bibliometric review was to identify and compare published research about selected infections and infectious diseases of Arctic animals in the circumpolar north. Our review utilizes research indexed in larger bibliographic databases (PubMed, Scopus, and eLibrary.ru) and covers infections in selected Arctic mammals, birds, and fish, looking for geographic trends, knowledge gaps, and emerging infections.

## 2. Materials and Methods

### 2.1. Eligibility Criteria

Our review collected information from publications on infections/infectious diseases in selected Arctic wildlife. The inclusion criteria that articles needed to meet were as follows: (1) data must come from the Arctic, defined according to the borders of the map “Arctic administrative areas” (Figure 2), (2) data must consist of original research or surveillance reports, (3) data must be published between 1990 and 2018, and (4) data must relate to occurring infections in the selected Arctic wild animals. Review articles, experimental infection studies, or studies on captive populations were excluded.

The following mammals were selected for this review: reindeer/caribou (*Rangifer tarandus*), moose (*Alces alces*), wolves (*Canis lupus*), sled dogs (*Canis lupus familiaris*), bears (*Ursus* spp.), foxes (*Vulpes* spp.), and hares (*Lepus* spp.). Selected birds included gulls (Laridae spp.), geese and ducks (Anatidae spp.), and grouse (Tetraoninae spp.), and selected fish included salmon and trout (Salmoninae spp.), cod (*Gadus* spp.), pike (*Esox* spp.), and cyprinids (Cyprinidae spp.). These wildlife animals were selected because of their importance to the Arctic environment and their close relationships with humans. Sled dogs and reindeer, although domestic and semidomestic, respectively, were included because of their integral position to human activities in the Arctic.

Because of varying definitions of “Arctic” and the migratory patterns of some animals, we included articles only if they focused on data from the Arctic provinces of Arctic countries, such as the United States (Alaska), Canada (listed further), Denmark (Greenland), Iceland, Norway (Troms, Finnmark, Nordland), Sweden (Norrbotten, Västerbotten), Finland (Northern Ostrobothnia, Kainuu, Lapland), northern Russia (listed further), and Northern Atlantic, Northern Pacific, or the Arctic Ocean. The following provinces from the Arctic zone of Russia were included: Murmansk Oblast, Republic of Karelia, Arkhangelsk Oblast, Nenets Autonomous Okrug (AO), Komi Republic, Yamalo-Nenets AO, Khantu-Mansi AO, Krasnoyarsk Krai (including only the northernmost Taimyr Dolgano-Nenets area), Republic of Sakha (Yakutia), Kamchatka Krai, and Chukotka AO. Canadian provinces with Arctic territories—Yukon, Nunavut, Northwest Territories, Labrador, and Newfoundland—were included, and southern provinces were considered on a case-by-case basis, provided the research had significance for the Arctic (particularly animals with ranges that extend into the southern regions of Canada). Similar selections were made for Fennoscandian studies (Norway, Sweden, Finland). In the eLibrary.ru search, we manually excluded all studies that were not relevant to one or several Arctic provinces of Russia.

Searches for English literature were conducted using PubMed and Scopus. Searches for Russian literature were conducted using eLibrary.ru. Only articles published since 1990 were included because eLibrary.ru started a systematic collection at that time with full texts, and we could have a more even comparison of published research papers in all three databases. In eLibrary.ru, the searches were restricted to research articles, books, and dissertations written in Russian. Included articles needed to have an electronic copy or print copy provided by the University of Oulu or via eLibrary.ru registered customer access.

### 2.2. Search Strategy and Inclusion Criteria

The objective of this bibliometric review was to identify selected infections and infectious diseases of Arctic animals in published literature.

In the Pubmed database, we used the MeSH term “animal diseases” to identify infectious diseases because it was a more all-encompassing term than specific infections. For article searches in PubMed and Scopus, the following search terms were used for mammals, including synonyms and scientific names joined by Boolean operators: “reindeer or caribou or rangifer tarandus”, “moose or alces alces”, “sled dog or sledge dog”, “brown bear or ursus arctos”, “black bear or ursus americanus”, “polar bear or ursus maritimus”, “arctic fox or vulpes lagopus”, “red fox or vulpes vulpes”, “wolf or canis lupus”, “hare or lepus”. In English language databases, the search terms for birds included: “avian influenza”, “gull”, “duck”, “goose”, “geese”, “grouse”, “anas”, “anser”, “chen”, “branta”, “somateria”, “polystica”, “clangula”, “melanitta”, ”larus”, “phasianidae”. For searches for fish, the following English search terms were used: “salmon or trout or salmonidae”, “cod or gadidae”, “pike or esocidae”, “cyprinidae or minnow”, “anisakidae or diphyllobothrium or opisthorchis”. We included genus and family level search terms for fish and birds because of their wide range of species.

We recognize that there are many more animals which are important to the Arctic ecosystem, human life, and health, but they were determined to be beyond the scope of this review. Every search involved the following geographic terms: Arctic regions, subarctic, polar, circumpolar, Alaska, Canada, Canadian Arctic, Greenland, Iceland, Norway, Svalbard, Faroe Islands, Sweden, Finland, Lapland, Scandinavia, Fennoscandia, Russia, Russian Arctic, Siberia.

The following term combinations were searched in eLibrary.ru: “Арктика” (Arctic), “северный” (northern), “инфекциoнные бoлезни живoтных” (infectious diseases of animals); “инфекции живoтных” (infections of animals), “инфекциoнные бoлезни млекoпитающих” (infectious diseases of mammals), “инвазивные бoлезни живoтных” (invasive diseases of animals), “инвазии живoтных” (invasions of animals), “инвазивные бoлезни млекoпитающих” (invasive diseases of mammals), “инвазии млекoпитающих” (invasions of mammals), “млекoпитающие” (mammals), “живoтные” (animals), “oлень” (reindeer), “медведь” (bear), “лисица” (fox), “заяц” (hare), “вoлк” (wolf), “ездoвая сoбака” (sled dog), “трихинелла” (trichinella), “тoксoплазма” (toxoplasma), “сибирская язва” (anthrax), “туляремия” (tularemia), “другие инфекции” (other infections), “инфекции в дикoй прирoде” / “инфекции в живoй прирoде” (infections in wildlife), “инфекции в живoтнoм мире” (infections in animal world), “инфекции фауны” (infections in fauna). The following search terms for birds were used: “птичий грипп” and “грипп птиц” (avian influenza) and many more for fish: “вирусные бoлезни рыб” (viral diseases of fish), “инфекциoнные бoлезни рыб” (infectious diseases of fish), “бактериальные бoлезни рыб” (bacterial diseases of fish), “паразиты рыб” (fish parasites), “инфекции рыб” (fish infections), “гельминтoзы рыб” (fish helminthoses), “инвазии рыб” (fish invasions), “лoсoсь паразиты” (salmon parasites), “лoсoсь инфекции” (salmon infections), “фoрель паразиты” (trout parasites), “фoрель инфекции” (trout infections), “щука паразиты” (pike parasites), “щука инфекции” (pike infections), “треска паразиты” (cod parasites), “треска инфекции” (cod infections), “гoрбуша паразиты” (humpback salmon parasites), “гoрбуша инфекции” (humpback salmon infections), “карпoвые паразиты” (carp parasites), and “карпoвые инфекции” (carp infections). In eLibrary.ru, only one search term or phrase can be searched at a time. As a result, there were many searches with overlapping results and nonrelevant articles.

During the search process, article titles and abstracts were evaluated for their relevance to the inclusion criteria. Articles that possibly met the inclusion criteria were then selected for full text review. Original studies and surveillance reports presenting the abovementioned search terms and prevalence or incidence calculations were included in the review.

All the search results were pooled, and duplicates were removed using Zotero (Corporation for Digital Scholarship, George Mason University, Fairfax, VA, USA) and RefWorks (Ex Libris, a ProQuest company, Jerusalem, Israel) software. Appendix A is provided in the Appendix A “Search terms used”.

### 2.3. Data Extraction

From the retrieved articles, the following information was recorded: article titles, publication year, country of research, and region, province, city, town if available, animals studied (species, sex, and age when relevant), type of infection, objectives of the study, and main results of the study. The collected data enabled analysis of the number of publications by country/region, infection, and animal species.

The authors recognize that it is not possible to perform prevalence studies on infections leading to rapid death of the infected animal, and prevalence studies are best suited for infections with low pathogenicity leading to chronic infection in animals.

### 2.4. Statistical Analysis

Our data analysis involved both quantitative (i.e., frequencies and percentages) and qualitative (i.e., thematic analysis) methods. In the quantitative analysis, frequencies and percentages were presented to summarize the bibliometric data. We used the chi-squared test to evaluate the statistical significance of differences between bibliographic databases in distributions of host type. The findings were also illustrated by graphic presentations. Linear trends in article publications from 1990 to 2018 were estimated using linear regression lines. We report a general bibliometric summary of the included articles first, followed by summaries of the main findings of the evaluated studies on infection diseases.

## 3. Results

### 3.1. Bibliometric Summary of Included Studies

We identified 4919 records within the three electronic databases (Figure 3). A total of 3368 articles were found in eLibrary.ru, 732 articles were found in PubMed, and 819 articles were found in Scopus. There were 704 duplicates that were removed from the English language databases. After the search results were reviewed for their relevance based on title and abstract, 552 articles were selected for a full text review. Of them, 200 were excluded and the remaining 352 articles were selected to be included in this review. Of the selected articles, 122 articles (35%) were found via eLibrary.ru and 230 articles (65%) were collected via PubMed and Scopus. Common reasons for exclusion included experimental studies, studies not giving data on distribution of the infection, articles in languages other than English and Russian, and articles not relevant to wildlife infections in the Arctic area. A detailed list of the included articles is provided in the Appendix A. The Appendix A includes the basic characteristics of all reviewed articles.

### 3.2. General Bibliometric Findings

From the total of 352 articles, there were 231 about mammals, 79 about fish, and 42 about birds (Table 1). More articles about mammals were selected from PubMed and Scopus than eLibrary.ru (*p* < 0.001). More studies about fish were selected from eLibrary.ru than PubMed and Scopus.

Reindeer/caribou were the most researched single host mammal both in the English and Russian databases, comprising 35% of all studies on mammals (Figure 4). Almost a quarter of the articles reported findings from more than one mammal species.

The articles included in this review were published in 115 journals. The 230 articles from the PubMed and Scopus databases were published across 77 journals. Table 2 displays the 20 most productive English language journals in terms of the number of publications selected. These journals published a total of 160 papers, accounting for 69.6% of the total number of English language articles. Most articles were published in the Journal of Wildlife Diseases (n = 48, 20.9%), Veterinary Parasitology (n = 17, 7.4%), Parasitology Research (n = 13, 5.7%), Journal of Parasitology (n = 10, 4.3%), PLoS ONE (n = 9, 3.9%), Parasitology (n = 8, 3.5%), Veterinary Record (n = 7, 3.0%), and Parasites and Vectors (n = 6, 2.6%) (Table 2). The other journals had 1–5 included articles in this review. A full record of the journals indexed by PubMed and Scopus can be found in Appendix A.

The 122 Russian articles included were published across 60 journals, with the most articles coming from Паразитoлoгия (Parasitology) (n = 17, 13.9%), Рoссийский паразитoлoгический журнал (Russian Parasitological Journal) (n = 7, 5.7%), and Прoблемы oсoбo oпасных инфекций (Problems of Particularly Dangerous Infections) (n = 7, 5.7%). A full list of Russian journals is included in Appendix A.

Publication trends are illustrated in Figure 5. Publications related to Arctic infections in wildlife (from our selected articles) have increased over time, with publications about mammals having the largest increase in numbers of articles published in all databases (PubMed, Scopus, and eLibrary) (*p*-values for linear increase trends are as follows: <0.001 (all articles), <0.001 (mammals), 0.001 (fish), 0.042 (birds)). Publication trends divided by English and Russian databases can be seen in the Appendix A.

The majority of the research in the English language databases was from Norway (n = 59, 16%), Canada (n = 40, 11%), Alaska (n = 39, 11%), and Finland (n = 29, 8%), but also covered all other Arctic countries. When including publications from English and Russian databases, most publications focused on the Russian Arctic (n = 131, 37%), of which 122 were found from the eLibrary.ru database (Figure 6).

The frequently published infections in mammals were *Trichinella* spp. (n = 39), *Brucella* spp. (n = 25), *Bacillus anthracis* (n = 25), rabies virus (n = 11), *Francisella tularensis* (n = 9), and *Echinococcus* spp. infection (n = 10) (Appendix A). In Russian, there were 25 articles on anthrax, while there were none found in English. There were nine articles on *Echinococcus* infections; two of those nine were found in the Russian database and the rest in the English language databases.

A total of 79 articles covered fish infections—34 in English and 45 in Russian. The infectious agents in the reviewed English language papers were parasites, especially anisakid nematodes, but also, e.g., *Diphyllobothrium* spp. cestodes (Appendix A). The distribution of infectious agents in Russian fish studies was as follows: parasites 39, bacteria 3, multiple 3. The majority of the Russian research articles focused on helminths (n = 40).

### 3.3. Results of Individual Studies

In the next section, we highlight those zoonotic infectious agents among mammals that are important for human health such as *Trichinella* spp., *Brucella* spp., *Bacillus anthracis*, rabies virus, *Francisella tularensis*, and *Echinococcus* spp. In addition, emerging, apparently nonzoonotic ones such as, e.g., *Setaria tundra* and the prions causing chronic wasting disease are also highlighted. Infections of herpesvirus, *Diphyllobothrium* spp., and canine viruses were also high in number of publications but are not discussed in detail in the review (they can be seen in Appendix A). Next, we discuss avian *influenza A virus* causing avian influenza in birds (n = 42 papers), and parasites and infectious diseases in fish (n = 79), part of which was focusing mainly on the aquatic ecology of fish populations (beyond the scope of this paper), and another part was about the prevalence and diseases in fish (more in the Appendix A).

#### 3.3.1. Infections of Mammals

There were 231 articles focusing on mammals; 64 Russian and 167 English publications selected for full text analysis (see Table 1 and the Appendix A). Zoonotic infections are of particular concern for Arctic wildlife [20]. Our search revealed that most publications focused on the following zoonotic mammal infections: viruses—rabies virus (n = 11 publications); bacteria—*Brucella* (25), *Francisella tularensis* (9), *Bacillus anthracis* (25); and parasites—*Trichinella* (39) and *Echinococcus* (10). These infections have the potential to kill not only animals but also people, and they are all among the selected potential climate-sensitive infectious diseases (CSIs) relevant for Northern regions that have wildlife as intermediate host, vector, amplifier, or reservoir [20,21,22,23].

In each section, we mention in parentheses how many publications were focused solely on this particular infection and how many focused on multiple infections (x one infection/y multiple infections).

Trichinella. Trichinellosis (trichinosis, *Trichinella* infection) is an infection caused by nematodes of the genus *Trichinella*. There are currently eight recognized species and an additional five identified but unnamed genotypes [24]. In the muscles of the host, larvae can live dormant for years. In the Arctic, an important survival factor for the parasite is resistance to freezing, which enables transmission to carrion feeders even from carcasses of animals that died during the winter. Three of the thirteen species/genotypes known are considered especially freeze-resistant, and their known distribution is limited to the Arctic and subarctic [24,25,26]. Altogether, there were 39 articles included about *Trichinella* spp. infections: 17 English (13 studying *Trichinella*/4 multiple infection papers—investigating it together with other infections) and 22 Russian (19/3). In the English language databases, the articles were published between 1990 and 2018, and in the Russian language database between 2007 and 2017.

Brucella. Brucellosis is a bacterial infection caused by the Gram-negative *Brucella* spp. and can occur in both humans and animals. There are 10 known *Brucella* species globally from the tropics to the Arctic [27]. Especially in the Arctic, where raw or inadequately cooked meat is traditional food, eating infected meat is the most common route of transmission, especially to Indigenous peoples. In the Arctic, *Brucella* spp. are found both in marine and terrestrial mammals. In humans and wild and domestic mammals, *Brucella* spp. infections are rarely fatal, but can cause severe symptoms including mastitis, abortion, orchitis, and osteomyelitis [28,29].

Altogether, there were 25 articles about *Brucella* spp. infections: 13 from English databases (9/4, respectively) and 12 from eLibrary.ru (6/6). The English papers were published between 1993 and 2018, and Russian ones between 2011 and 2018.

Rabies virus. Rabies is caused by lyssaviruses maintained primarily in bats (Chiroptera spp.) and foxes [30,31]. In bats, many different lyssaviruses are known in different parts of the world, often with rather mild pathogenicity [32]. Rabies causes inflammation of the brain, which is considered fatal. The infection is spread via the bite of an infected mammal. The incubation time before the onset of symptoms may be long, even many months, depending on the distance of the bite site from the brain.

There were 11 articles that elaborated on the rabies epidemiological situation in the Arctic: seven English (4/3) and four Russian (1/3). The English language papers spanned from 2001 to 2014, whereas the Russian papers were more up to date, 2011 to 2018.

Francisella tularensis. Tularemia (*Francisella tularensis* infection) is considered peracute, plague-like, and highly fatal to leporids, such as hares, and knowledge about this infection in wildlife is usually based on necropsy of animals found dead or dying [33]. However, when infected orally, rodents have been suggested to become chronically infected by *F. tularensis* and shed bacteria [34]. The causative agent is a Gram-negative coccobacillus. Humans are mainly infected via arthropod vectors or via direct contact with infected animals, though aerogenic transmission is also possible. There were altogether nine papers about *Francisella tularensis*: three in English (1/2) and six in Russian (5/1), published in 1998–2018.

Bacillus anthracis. Anthrax is caused by the Gram-positive aerobic rod *Bacillus anthracis*. The infection is often regarded as peracute, leading to sudden death of ruminants [35]. Humans can also become seriously ill, and the bacterium has been studied as a biological weapon [35]. *Bacillus anthracis* is dangerous for its ability to form spores that can survive in the soil for decades, and possibly for much longer in permafrost [35]. Anthrax is commonly studied in the Russian Arctic using a descriptive approach; the historic spread of this disease is known as dangerous and was common in pre-1917 Russia (e.g., [36]), as well as in recent emergency situations (e.g., [37]). In the summer of 1911, 100,000 reindeer were reported to have succumbed to anthrax on Yamal Peninsula (see [38]). Anthrax is also called Siberian plague and, nowadays, is most relevant in Sakha Republic (Yakutia), Yamal, and Taimyr Peninsula.

There were only 3 review papers about anthrax found in English that were not included, because they did not meet the inclusion criteria. In our review, 25 (17/8) Russian publications were included mainly concerning the infection in Yamal Peninsula and published between 2011 and 2018. Seventeen papers focused on ecological, economic, and societal consequences and various regional features of a sole infection—anthrax. The other eight papers investigated multiple infections, anthrax being one among several others regarding prognostic modelling of infections and veterinary provisions for northern reindeer and agricultural science.

Echinococcus. Echinococcosis or hydatid disease is a zoonotic infection caused by the larval forms of the taeniid tapeworms of the genus *Echinococcus*. There are several species of *Echinococcus* with different intermediate host species, but most *Echinococcus* species have canids as the definitive host. The definitive host usually does not suffer from the small (2–5 mm) adult tapeworms in the intestine, but the intermediate host develops cysts in tissue, most often the liver or lung, and may become more prone to be eaten by the definitive host [39,40]. Cystic echinococcosis in humans is caused by the *Echinococcus granulosus* complex (*E. granulosus sensu lato*), whereof the cervid genotypes *Echinococcus canadensis* G8 and G10 (genotypes 8 and 10) are typically Arctic and boreal in distribution having moose, deer (*Odocoileus* spp.), and reindeer/caribou as natural intermediate hosts, and humans as accidental aberrant hosts. Humans can become infected by ingesting worm eggs originating from the feces of the definitive host, often the domestic dog. *Echinococcus multilocularis* causes alveolar echinococcosis in the human liver, but its natural life cycle rotates between foxes as definitive hosts and small rodents, usually voles (Arvicolinae spp.), as intermediate hosts [39]. Human cystic echinococcosis in the Arctic caused by *E. canadensis* G10 is usually a mild infection not requiring radical therapy, but alveolar echinococcosis was, before benzimidazole chemotherapy, very often fatal [40,41]. The WHO considers echinococcosis a neglected (tropical) disease [42].

There were 10 papers considered in this section, 7 relevant English language papers reported from the Arctic areas of Canada, Finland, Norway, Sweden, and Russia (6/1) and from eLibrary.ru (0/3). The dates of the English language papers ranged from 1998 to 2014, and the studies related to multiple animals including arctic and red fox, wolf, reindeer, moose, and dog.

#### 3.3.2. Emerging Diseases in Mammals

Chronic Wasting Disease. CWD is a fatal contagious neurodegenerative prion disease in cervids. Prion diseases are caused by a conversion in the cellular prion protein PrP^C^ into the pathogenic PrP^Sc^. PrP^Sc^ accumulates in the brains of infected animals, resulting in fatal neurodegeneration. Species-specific polymorphisms in the prion protein gene *Prnp*, however, determine susceptibility to CWD [43]. Since the 1960s, CWD has been detected in 23 US states and 2 Canadian provinces [43].

CWD is dangerous to Arctic wildlife but not yet widely reported in the Arctic region during the time frame of our searches in English or Russian language databases. There were only two English language publications on CWD from non-Arctic Canada and Norway; these full texts were relevant to this review as they study the northern animals—caribou and reindeer. Only one article mentioning CWD was found in the eLibrary.ru database, and no cases have been registered in Russia so far [44].

Setaria tundra. Coinciding with decades of warming and anomalies of high temperature in the subarctic region of Fennoscandia, the mosquito-borne filarioid nematode *S. tundra* is now associated with an emerging epidemic disease resulting in substantial morbidity and mortality for reindeer and moose [45]. The infection causes peritonitis and perihepatitis, which cause significant economic losses due to reduced body weight of infected animals [46]. Increasing temperatures may facilitate a range expansion and increasing duration of the effective transmission period for *S. tundra* [46]. In our review, there were four English language papers including *Setaria* spp. published between 2007 and 2018.

#### 3.3.3. Infections of Birds

Avian Influenza. Avian influenza (AI) is a disease of wild and domestic birds, and occasionally also humans and other mammals. It is caused by Avian influenza viruses belonging to the Orthomyxoviridae family and in the genus *Influenza virus* A [47]. Two surface proteins—hemagglutinin and neuraminidase—are used for subtype classification of *influenza A virus*. To date, 16 hemagglutinins (H1 to H16) and 9 neuraminidases (N1 to N9) subtypes have been reported. The AI viruses are classified into low (LPAI) and high pathogenic (HPAI) strains, based on their pathogenicity to laboratory chicken. Both LPAI and HPAI circulate in wild birds, especially in waterfowl, gulls, and shorebirds. All HPAI strains isolated so far have been either H5 or H7, but most H5 or H7 subtypes isolated have been LPAI. Pathogenicity to various wild bird species may differ considerably from that of chickens [47]. Birds have the potential to spread certain diseases to endemic Arctic species or to even spread disease agents such as H5N1, a highly infectious influenza strain, to humans [48].

Seventy English and twenty-five Russian papers on wild bird avian influenza were initially selected from the databases as possibly relevant to the review based on their title and abstract. Twenty-nine English publications (2000–2018) and thirteen Russian papers (2001–2016) were included in this review after full text analysis (Appendix A). The main criterion for inclusion was a sampling of birds from one or several Arctic locations. There was a case-by-case assessment of the studies where samples were taken in the southern provinces of Arctic countries. If the sampling region was stated and all or some birds were Arctic resident or breeding migratory species, the papers were included. If the sample location was unknown, the birds were not Arctic/sub-Arctic species, and/or their typical breeding ground was outside the Arctic, the papers were excluded. If available, the season of sampling was additional information on which we based our decision to include or exclude the study. A large number of Ottenby Bird Observatory studies in Sweden were excluded (n = 11) as the samples contained a low proportion of migrating ducks that come from the Arctic. However, those studies provide a wide range of information on *influenza A virus* in wild birds migrating in northern Europe.

#### 3.3.4. Infections of Fish

Wild and farmed fish can be sources of a variety of zoonotic parasites. Arctic marine fish can harbor anisakid nematodes, having cetaceans or pinnipeds as natural final hosts. Moreover, they can host larval stages of *Diphyllobothrium* spp. cestodes, with marine mammals as natural final hosts. Freshwater fish can harbor other *Diphyllobothrium* species with fish-eating mammals or birds as final hosts. Freshwater fish may also have metacercaria larvae of opisthorchiid liver flukes. Adult flukes reside in the bile ducts of piscivorous mammals or birds. In our review (n = 79), there were two different types of publications concerning fish parasites and infectious diseases, one part focusing mainly on aquatic ecology of fish populations (n = 39), and rest focusing on the prevalence of parasite infections and diseases in fish (18 English, 22 Russian papers) (Appendix A).

## 4. Discussion

### 4.1. General

Assessing wildlife infection trends in the changing Arctic poses many challenges. Wildlife in the Arctic are marine, freshwater aquatic, terrestrial, and aerial, spanning eight countries or more, if the species are migratory. While more research is needed to address all the factors affecting regional wildlife health, the intention of this paper is to increase understanding of the status of knowledge of selected infections in Arctic wildlife. Primarily, we wanted to review recent, prepandemic research on infections in Arctic wildlife in English and Russian databases in order to identify knowledge regarding existing and emerging infections.

Our review is apparently the first bibliometric review to provide an overview of the status of research on infections in Arctic wildlife that includes Russian literature. The number of published papers in English and Russian have increased 7–10-fold in all categories (mammals, fish, and birds) since 1990. However, the increase in the number of papers concerning avian influenza started rather late, in 2005. In our review, 352 articles were selected to be included, and among this number, 122 articles (35%) were found via the Russian database eLibrary.ru. Most research features Russia (37%), Norway (16%), Canada (11%), Alaska (11%), and Finland (8%). It is important to include Russian articles, allowing non-Russian readers some access to what is published in Russian scientific literature [19].

Reindeer husbandry is an integral part of the Arctic and is practiced mostly in the Sápmi homeland of northern Fennoscandia and the Arctic Russia [49], and there were altogether 81 articles concerning infectious diseases in reindeer or caribou. While the Fennoscandian reindeer have been spared from brucellosis and anthrax, these bacterial infections have been a nuisance in Russia, where vaccination of reindeer against anthrax was widely practiced during the Soviet era, but not during the last decades [50]. The processes of permafrost thaw, overgrazing, and political interference with herding practices have released *B. anthracis* spores and caused new anthrax outbreaks.

There is a greater risk of zoonotic infections in the Arctic now than earlier [11,22]. Several zoonotic infections in the Arctic are of special concern and found in all investigated databases, such as viruses causing avian influenza and rabies, bacteria such as *Brucella* and *Francisella*, and the parasites *Trichinella* and *Echinococcus*. There were other infections studied extensively only in the Russian literature (bacterium *Bacillus anthracis*) or in English literature (parasites *Toxoplasma gondii* or *Diphyllobothrium* spp., canine viruses, and herpesvirus). Other infectious agents such as *Cryptosporidium*, *Giardia*, *Hypoderma*, *Marshallagia*, *Ostertagia*, *Salmonella*, and *Morbillivirus* were less frequently studied. The research interests vary by longitude. For example, research in Alaska focused heavily on viruses (51% of all papers included in the study), while Finland, Greenland, and Norway have investigated mostly parasites (about 70% of all included papers per country). Bacteria in animals were studied in every Arctic country—comprising as low as 8% of included papers about Alaska and Canada to the maximum of 28% of papers about Russia. Overall, pan-Arctic guidelines are needed for monitoring and surveillance in wildlife, especially for these included infections, which have a significant impact for humans. The importance of zoonotic diseases is clearly seen with the ongoing COVID-19 pandemic.

Increased globalization and human interest in the Arctic bring additional challenges that also impact infectious diseases among wildlife. The rapid influx of people and pets has the potential to introduce new infections, establish new hosts, and alter the environment through industry, construction, and pollution. Additionally, the changes can stress animals, making them more vulnerable to infectious diseases [48]. All of these factors contribute to increasing the risk of infectious diseases to Arctic wildlife and humans [45,51,52], and programs to continuously monitor wildlife infections are urgently needed.

### 4.2. Future Directions

The aim of our bibliometric review was to provide descriptive coverage of the published literature regarding the selected infectious diseases/infections in wildlife during the last decades. The strength of our review is in identifying areas where opportunities for further research, collaboration, systematic reviews, or monitoring programs are possible and needed.

#### 4.2.1. Pathogen Mutation and Host Switch Research

There are several hitherto neglected opportunities for research on zoonotic wildlife infections and those otherwise important for human welfare and food security in the Arctic. The scientific community should be alerted to emerging and re-emerging infections. There is limited scientific evidence on infections traditionally known by Indigenous peoples, who may have learned to recognize the infection as part of their daily living and know how to avoid it [53].

Mutation and host switch of agents from animals to human host could be investigated more actively [54,55]. The world has been and is changing at a rapid pace, and a recent assessment of biomass on Earth estimated the wild mammal biomass to be 0.007 Gt C (gigatons of carbon), while that of humans was 0.06 Gt C and livestock 0.1 Gt C [56]. Therefore, any infective agent “looking for” host switch has higher chances in finding the new host in the abundant and ubiquitous humans or domestic animals than in another wild mammal species.

The COVID-19 pandemic and the most recent multicountry monkeypox outbreak are prime examples of wildlife-derived human infections. Even though they were preceded by previous zoonotic coronavirus epidemics or small-scale local monkeypox outbreaks, the global spread of these human infections was not really expected.

One of the recent studies has found that Nearctic zooparasitic nematode *Orthostrongylus macrotis* was in a Palearctic host—Taimyr wild reindeer [57]. Supposedly this lungworm was introduced to Russia with muskoxen delivered from North America in the 1970s. This may be a case of host switch (or even double host switch since muskox is not a typical host for this invasive parasite species). There is also a danger of the spread of emerging diseases, such as CWD, among wild and domestic reindeer in the European part of Russia, and there are new studies, e.g., on possible genetic resistance to CWD [58].

#### 4.2.2. Guidance and Unification of Methodology

The validity and practical utility of observational research depends critically on good study design, appropriate analysis methods, and high-quality reporting and data presentation [59]. In the reviewed literature, we found inconsistent use of terminology and unclear data presentation. The reporting of observational findings often exhibited serious shortcomings. An efficient way to help readers extract the necessary data is to develop guidance documents on data presentation that are disseminated to the research community at large. We need a much more structured framework in scientific reporting, which emphasizes that today’s scientific evidence is based on the synthesis of studies reporting findings with similar effect size measures [60].

The diagnostic sensitivity of the different methods varies greatly regionally, e.g., clinical or laboratory diagnosis, macroscopic or microscopic examination, direct methods to show the infective agent or indirect methods to show its effects (often antibodies) [61]. Metagenomic analysis of environmental samples may also facilitate analysis for unknown infections. The lack of common programs and research method standards complicates result comparison between countries, regions, and times. In this review, we could not do a meta-analysis of the existing studies, since very few papers met the necessary criteria. It is especially important to improve and standardize these methodologies in the future to enhance research.

Harmonization of studies is needed to develop common Arctic protocols to undertake studies on infection in wildlife, e.g., a protocol to measure the prevalence of zoonotic diseases. Good examples are the *Trichinella* protocols/guidelines of the International Commission on Trichinellosis that can be applied in all Arctic countries [51].

#### 4.2.3. Involvement of Stakeholders in Surveillance and Risk Assessment

As the Arctic continues to change at a rapid pace, detecting changes in wildlife is paramount. The results of this bibliometric review indicate the importance of including Russian research when assessing the state of infection research regarding Arctic wildlife. Further studies on surveillance, vaccination, and education for people interacting with wildlife and living in the Arctic will be crucial for mitigating the changes in wildlife infectious diseases associated with globalization and climate change. To provide future valuable data, there is a need to combine the knowledge of Indigenous peoples, citizen scientists, hunters, and governmental agencies.

Wildlife surveillance is important for detecting new diseases and forming risk assessments, which are vital to both human and wildlife populations [52]. The AMAP, the *Arctic Monitoring and Assessment Programme*, is working on the risk estimation of contaminants in the environment, humans, and wildlife. In the Sustainable Development Working Group (SDWG), there is one important ongoing project—One Arctic, One Health (since 2015) [18]—which exemplifies monitoring, e.g., zoonotic diseases in the circumpolar Arctic.

However, there are still limited surveillance systems or monitoring networks, veterinary care, and knowledge on animal behavior, all of which can pose challenges for assessing wildlife diseases. Establishing new research projects focused on zoonotic diseases, such as CLINF “Climate change effects on the epidemiology of infectious diseases and the impact on Northern societies” are urgent [55,60,61].

### 4.3. Limitations

It is probable that we missed relevant documents that would have contributed additional empirically derived knowledge on infections in Arctic wildlife. We recognize that studies were excluded if they were published in a journal that is not represented in the three databases we used. We searched the databases that focus on health and life sciences. Thus, the databases selected may have a bias toward zoonotic infections, given that they mainly support human medical research. However, the Scopus database also covers fields that span multiple disciplines. In addition, the Russian database eLibrary.ru covers Russian journals that are not indexed by the general international bibliographic databases.

Our search terms also limited what articles were selected. Because this is a review, not every relevant term was possible to search; therefore, MeSH terms were used, when possible, to encompass as many search options as possible. In eLibrary.ru, only one search term or phrase can be searched at a time. As a result, there were many searches that produced overlapping results and nonrelevant results. However, we tried to unify our search method for all included databases and translated the English terms into the Russian language keywords including all possible combinations of synonyms. The importance of careful consideration of search terms cannot be overemphasized when planning the follow-up review on Arctic wildlife diseases in the post-COVID-19 era.

Limitations in chosen infections, hosts, and prevalence measurement (biases against infections with high mortality) are obvious. We decided which articles to include based on the inclusion criteria, discussed the infections exhaustively, and omitted many of those that we ourselves considered most interesting. This was done to gain a wide spectrum of wildlife infection sufficing as examples from an even wider diversity. We also made judgements on study locations, study units and infections, and methods used. Due to inconsistent reporting and terminology, this was not always straightforward and may have resulted in inadvertent exclusions. In addition, in keeping with accepted review methodology, we did not appraise the methodological quality of the articles that were included in our extraction. This means that the characteristics extracted have not been considered in context to the study design or methodological rigor of the work.

## 5. Conclusions

Our paper offers a bibliographic overview of the state of research in select Arctic wildlife infections during the time period of 1990–2018 in the entire Arctic region. There are noticeable differences in the tradition of wildlife infection research in the English and Russian literature and by geographical area. Research seems mainly based on the economical and public health importance of various infections in the region, but funding possibilities also have an important role. It would be important to form a circumpolar observing and researching network of wildlife diseases (especially in zoonotic diseases) in close collaboration with the public health sector. All this suggests future directions for One Health research in the Arctic.

## Figures and Tables

**Figure 1 ijerph-19-11260-f001:**
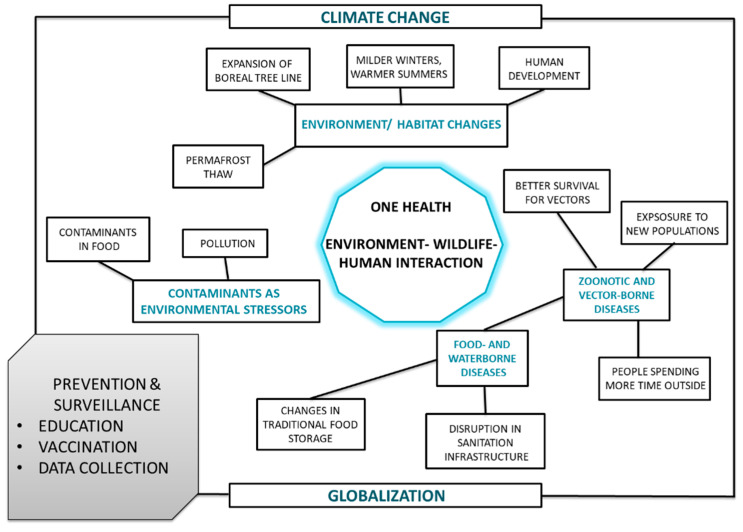
Environment—wildlife—human interactions in the context of climate change [2].

**Figure 2 ijerph-19-11260-f002:**
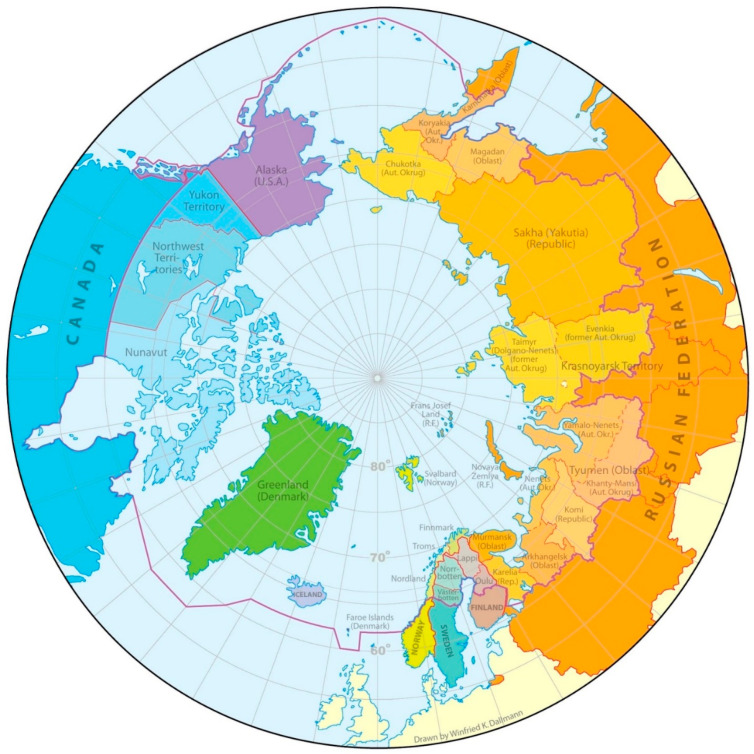
Arctic administrative areas. Map courtesy of Winfried K. Dallmann, Norwegian Polar Institute/Arctic Council.

**Figure 3 ijerph-19-11260-f003:**
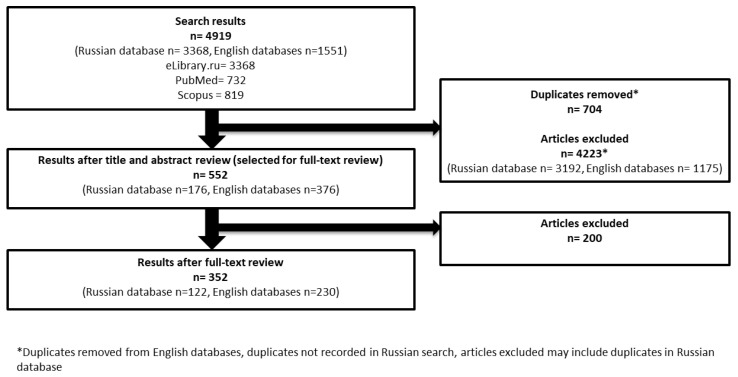
The review screening process: initial search results, articles selected for a full text review, and articles included in the review.

**Figure 4 ijerph-19-11260-f004:**
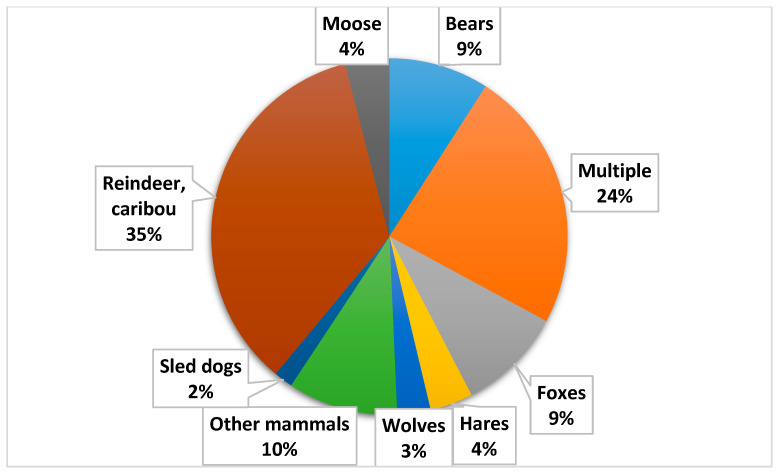
Percentage proportion of mammal species in the evaluated articles about the health status of Arctic mammals from 2009 to 2018 (n = 231).

**Figure 5 ijerph-19-11260-f005:**
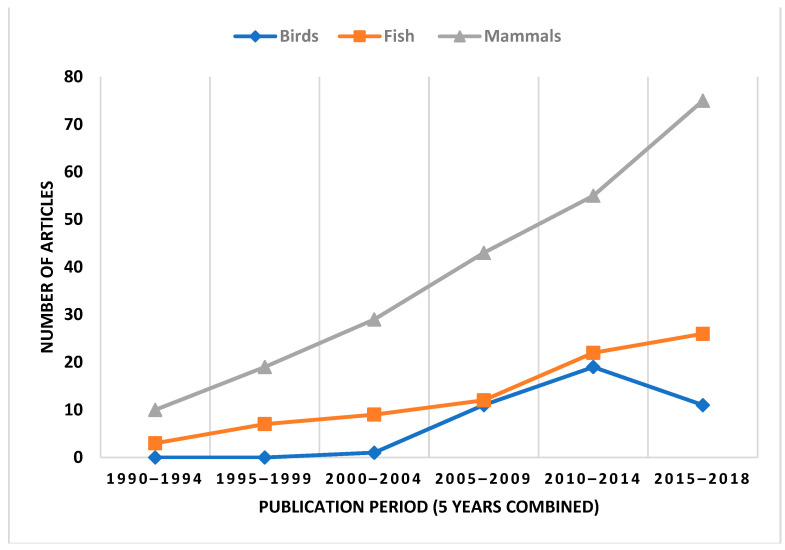
Trends in article publication used in this review from 1990 to 2018 divided by “mammals” (n = 231), “birds” (n = 42), and “fish” (n = 79). *p*-values for linear growth trends are as follows: <0.001 (all articles), <0.001 (mammals), 0.001 (fish), 0.042 (birds).

**Figure 6 ijerph-19-11260-f006:**
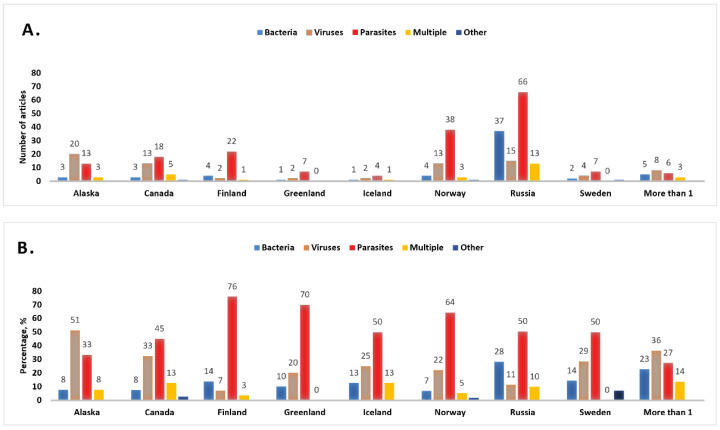
Types of infection in mammals and birds featured in articles by country of research in absolute numbers (**A**) and in percentage to the total amount of papers per country (**B**). “More than 1” refers to articles that have research in more than one country, and “Multiple” refers to articles that include more than one type of infection. “Other” refers to infections including prions, fungi, and arthropods (there were few publications in these areas, and they were therefore combined).

**Table 1 ijerph-19-11260-t001:** Frequency and percentage distributions of host type by bibliographic databases in the included articles (n = 352).

	Article Database		
	PubMed and Scopus	eLibrary.ru	Total
Host type	n (%)	n (%)	n (%)
Birds	29 (12.6)	13 (10.7)	42 (11.9)
Fish	34 (14.8)	45 (36.9)	79 (22.4)
Mammals	167 (72.6)	64 (52.4)	231 (65.6)
Total	230 (100)	122 (100)	352

**Table 2 ijerph-19-11260-t002:** Top English language journals, which published at least three articles on the selected wildlife infections/diseases in the circumpolar Arctic. The number of published articles is reported by the host type.

	Host			
Journal	Fish n	Bird n	Mammals n	All n
Journal of Wildlife Diseases	1	2	45	48
Veterinary Parasitology			17	17
Parasitology Research	4		9	13
Journal of Parasitology	3		7	10
PLoS ONE		6	3	9
Parasitology	1		7	8
Veterinary Record			7	7
Parasites and Vectors	1	1	4	6
Acta Veterinaria Scandinavica		1	3	4
Archives of Virology		4		4
Canadian Journal of Zoology	1		3	4
Diseases of Aquatic Organisms	4			4
Journal of Veterinary Diagnostic Investigation			4	4
Virology Journal		3	1	4
Avian Diseases		3		3
Emerging Infectious Diseases	1	1	1	3
IJP: Parasites and Wildlife			3	3
Journal of Fish Diseases	3			3
Oikos	1	1	1	3
Polar Biology			3	3

## Data Availability

Not applicable.

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
