# Peer review of "Research on Selected Wildlife Infections in the Circumpolar Arctic—A Bibliometric Review"

_ijerph, 2022, doi:10.3390/ijerph191811260_

Round 1

Reviewer 1 Report (Previous Reviewer 1)

Dear authors,

In my opinion you manuscript can be accepted for publishing in current form.

Reviewer 2 Report (Previous Reviewer 3)

You have made a good job

I wish you all the best for your research

This manuscript is a resubmission of an earlier submission. The following is a list of the peer review reports and author responses from that submission.

Round 1

Reviewer 1 Report

Dear authors,

Congratulations for the study presented here.

I find this topic important, and the idea of inclusion of literature in Russian language is great, but I have several major comments in relation to this manuscript.

  1. The body of the manuscript is quite big, but seems without good reason. There are a lot of sentences such as ,,Wild and farmed fish are good food for humans, but also eaten by other mammals, and fish can be sources of a variety of zoonotic parasites.'' You could just shorten it to - fish can be sources of a variety of zoonotic parasites, the rest is commonly known (humans and animals eat fish). So please, reduce the body of the manuscript by removing  commonly known facts (about 1/6 of the manuscript).
  2. In relation to discussion, you mostly addressed data as data, and the topic of wildlife infections is left on the side. I would like to see more focused discussion that is related to the pathogens under One Health tenets (that you mentioned straight away). In my opinion, discussion needs to be the backbone for this type of paper.
  3. I missed what is the One Health approach in this paper, could you please explain? This paper is related only to wildlife, where are the human cases?

Minor comments:

When using abbreviations such as COVID19, you firstly need to write the full name. 

Author Response

Dear Reviewer,

Thank you for your valuable comments. Below please find our answers to each comment. In additional, all changes should be visible in the revised manuscript, where we used track change mode to show all changes.

Comment 1: The body of the manuscript is quite big, but seems without good reason. There are a lot of sentences such as ,,Wild and farmed fish are good food for humans, but also eaten by other mammals, and fish can be sources of a variety of zoonotic parasites.'' You could just shorten it to - fish can be sources of a variety of zoonotic parasites, the rest is commonly known (humans and animals eat fish). So please, reduce the body of the manuscript by removing  commonly known facts (about 1/6 of the manuscript).

Response 1: The example sentence has been shortened accordingly. We have made several reductions to the manuscript, e.g. by deleting sub-chapters 4.2 and 4.3, and writing a new sub-chapter of 4.2. Future directions. There are also other parts in the manuscript where we removed text.

Comment 2: In relation to discussion, you mostly addressed data as data, and the topic of wildlife infections is left on the side. I would like to see more focused discussion that is related to the pathogens under One Health tenets (that you mentioned straight away). In my opinion, discussion needs to be the backbone for this type of paper.

Response 2: This was done since our paper is a bibliometric review, which is focused on particular infections, e.g. on Trichinella, Brucella, Rabies virus, Francisella tularensis, Bacillus anthracis, Echinococcus, and emerging diseases in mammals e.g. Chronic Wasting Disease, Setaria tundra, plus Avian Influenza in birds in detail. We have highlighted the One Health approach in our new sub-chapter 4.2 Future directions and Conclusions.

Comment 3: I missed what is the One Health approach in this paper, could you please explain? This paper is related only to wildlife, where are the human cases?

Response 3: Thank you very much for the important comment concerning One Health. The One Health approach is rather new platform in the Arctic, and it has been very recently used e.g. in the projects of the Arctic Council. In our bibliometric review we focus on published papers about selected wild-life infectious diseases and most of them are zoonotic. Our paper gives basic information about what has been studied in the circumpolar Arctic since 1990. In the sub-chapter of 4.2 Future directions, we point out the importance of connecting all aspects of One Health together: humans, wild-life and environment.  We have published a paper about infectious diseases in humans and climate change, see DOI: 10.1016/j.envint.2018.09.042.

Comment 4: Minor comments: When using abbreviations such as COVID19, you firstly need to write the full name. 

Response 4: Coronavirus disease 2019 added at a first instance.

Reviewer 2 Report

The topic of the review seems interesting, however it is also unclear what kind of comparison the authors aimed to conduct. The authors provide mostly numbers, rarely any comparable parameters like %. There is also no statistical analysis applied, thus I am not convinced that it is justified to write about ‘higher’ or ‘lower’ numbers: significance of difference between the numbers is not evaluated.

Main comments:

  • Lack of statistical analysis of the results – as specified above; the authors shall try to calculate more comparable parameters (%) and to compare them statistically (i.e. by Chi-square test), for example data from table 1 – numbers look quite similar, and it is difficult to evaluate if the numbers from Eng and Rus databases are significantly different. I am not sure what kind of analysis in mentioned in lines 195-196 (frequencies and percentages’).
  • I have spent some time to find that the authors were interested only in AIV infection n birds: this issue shall be clearly stated in Results, because this is responsible for the lowest number of selected papers on birds (line 223): it is also different selection criterium than for the other groups of animals (as I understood, all type of infectious agents were of interest for mammals and birds). Author may provide more detail explanation for the choice of AIV as the only selected pathogen in birds.
  • I am not sure if the same terms were used to search English- and Russian-language data bases. I noticed that this point is tackled in limitation section, but I think it should be explained in Methods. Max efforts shall be done to unify search method for all included databases.
  • Data on fish parasites: I think that the authors shall put more effort in expanding this description in the main text; it is extremely shortened and not very informative, contains errors (English papers are not all on anisakids- they are all on parasites, as great majority of Russian ones are [I do not think the difference could be stat significant]). There is a supplementary file on fish papers, but it provides a different kind of description than the remaining parts of ms.
  • I really miss more detail presentation of findings in the selected groups of animals (presented on fig. 4): I think this would be more informative than presentation of very basic description ‘by main pathogen’ i.e Trichinella, Brucella. Especially as the authors try to discuss data on infectious agents in reindeer in Discussion without presenting it in the Results.
  • Generally, I think that Discussion is not focused on the present finding, contains a lot of statements without any support from scientific sources(references) and discuss problem not presented in review (climate warming – I can not see any analysis of the present data with climate change). Conclusions are not clear.
  • Section “knowledge gaps’ : I am not sure what is all about: it is very speculative, majority of statements seems authors’ opinion not facts supported either but their analysis from the current paper or by other papers. I do not find it appropriate to have such speculative chapter in scientific paper.

Minor issues:

English language shall be improved: for example in Abstract is ‘select’ shall be selected’

The first sentence form the Discussion has no sense (sth missing?)

Ms shall be checked and correct by English native speaker.

Francisella tularensis is not actually vectored by ticks (majority of recent studies found only F. tularensis-like endosymbionts in ticks).

Latin names of organisms shall be italicised, and genus name abbreviated after first use (please check part on Seratia tundrae).

As the author of the Fig 2 is not the co-author of the current ms, do the authors have permit to use it?

Citation: Publication trends divided by English and Russian databases can 257 be seen in S4 Fig.; in zip file there is no Fig S4

Fig 5 caption: please specify  ‘AIV in birds’

Lines 286-287: ‘The infectious agent was anisakidae parasite in all reviewed English language papers (S1 File). ‘ correct: parasites, also other the anisakidae, for example cestods (i.e. Diphyllobothrium)

Line 456-461: ? very basic

Section 4.3. Risk of bias – it is also rather speculative, without proper support from cited papers. Please, rewrite, add relevant citations to support your opinions.

Conclusions: ? very general, not focused on the present study, because some of the topics were not studied/analyzed  here.

Author Response

Dear Reviewer,

Thank you for your valuable comments. Below please find our answers to each comment. In additional, all changes should be visible in the revised manuscript, where we used track change mode to show all changes.

Reviewer 2 – Comments and Suggestions for Authors

English language and style

( ) Extensive editing of English language and style required
( ) Moderate English changes required
(x) English language and style are fine/minor spell check required
( ) I don't feel qualified to judge about the English language and style

The topic of the review seems interesting, however it is also unclear what kind of comparison the authors aimed to conduct. The authors provide mostly numbers, rarely any comparable parameters like %. There is also no statistical analysis applied, thus I am not convinced that it is justified to write about ‘higher’ or ‘lower’ numbers: significance of difference between the numbers is not evaluated.

Main comments:

Comment 5: Lack of statistical analysis of the results – as specified above; the authors shall try to calculate more comparable parameters (%) and to compare them statistically (i.e. by Chi-square test), for example data from table 1 – numbers look quite similar, and it is difficult to evaluate if the numbers from Eng and Rus databases are significantly different. I am not sure what kind of analysis in mentioned in lines 195-196 (frequencies and percentages’).

Response 5: We have now added more percentages to the text and table/figure captures in addition to the frequencies. Statistical significance of findings presented in Table 1 and Figure 5 are now evaluated using chi-square test and linear regression slopes.

Comment 6: I have spent some time to find that the authors were interested only in AIV infection n birds: this issue shall be clearly stated in Results, because this is responsible for the lowest number of selected papers on birds (line 223): it is also different selection criterium than for the other groups of animals (as I understood, all type of infectious agents were of interest for mammals and birds). Author may provide more detail explanation for the choice of AIV as the only selected pathogen in birds.

Response 6: Thank you for this comment. We have focused only on AIV, and this limitation is now stated more clearly in the 3. Results.

Comment 7: I am not sure if the same terms were used to search English- and Russian-language data bases. I noticed that this point is tackled in limitation section, but I think it should be explained in Methods. Max efforts shall be done to unify search method for all included databases.

Response 7: We have tried to use the same search terms as much as possible. However, every database has its own specific terms. In eLibrary.ru, it is possible to only use one search term at a time. We have added text in the Methods and Limitation sections.

Comment 8: Data on fish parasites: I think that the authors shall put more effort in expanding this description in the main text; it is extremely shortened and not very informative, contains errors (English papers are not all on anisakids- they are all on parasites, as great majority of Russian ones are [I do not think the difference could be stat significant]). There is a supplementary file on fish papers, but it provides a different kind of description than the remaining parts of ms.

Response 8: Thank you for your comment. We have expanded text in the fish section.

Comment 9: I really miss more detail presentation of findings in the selected groups of animals (presented on fig. 4): I think this would be more informative than presentation of very basic description ‘by main pathogen’ i.e Trichinella, Brucella. Especially as the authors try to discuss data on infectious agents in reindeer in Discussion without presenting it in the Results.

Response 9: We have added text about the different species in the Figure 4, especially those which have been in the published papers, e.g. reindeer.

Comment 10: Generally, I think that Discussion is not focused on the present finding, contains a lot of statements without any support from scientific sources(references) and discuss problem not presented in review (climate warming – I can not see any analysis of the present data with climate change). Conclusions are not clear.

Response 10: Thank you for your comment. The Discussion and Conclusion sections have been modified, and there are now more references.

Comment 11: Section “knowledge gaps’ : I am not sure what is all about: it is very speculative, majority of statements seems authors’ opinion not facts supported either but their analysis from the current paper or by other papers. I do not find it appropriate to have such speculative chapter in scientific paper.

Response 11: We have deleted the section of Knowledge gaps and modified a new section of 4.2. Future directions, where we focus on the role and impact of our bibliometric review and also suggest opportunities for future research. We hope this sub-chapter can be food for thought for establishing new research projects including all aspects of the One Health approach.

Minor issues:

Comment 12: English language shall be improved: for example in Abstract is ‘select’ shall be selected’

Response 12: A native English speaker has checked the revised version.

Comment 13: The first sentence form the Discussion has no sense (sth missing?)

Response 13: We changed the order of wording and hope it is more understandable.

Comment 14: Ms shall be checked and correct by English native speaker.

Response 14: A native English speaker has checked the revised version.

Comment 15: Francisella tularensis is not actually vectored by ticks (majority of recent studies found only F. tularensis-like endosymbionts in ticks).

Response 15: Thanks for the comment. We changed the text to “arthropod vectors”.

Comment 16: Latin names of organisms shall be italicised, and genus name abbreviated after first use (please check part on Seratia tundrae).

Response 16: Thank you; we have abbreviated it.

Comment 17: As the author of the Fig 2 is not the co-author of the current ms, do the authors have permit to use it?

Response 17: We have permission from the figure creator via email and have include his name and institution into the Figure 2 caption.

Comment 18: Citation: Publication trends divided by English and Russian databases can 257 be seen in S4 Fig.; in zip file there is no Fig S4

Response 18: Thanks for noticing this. We have added it.

Comment 19: Fig 5 caption: please specify  ‘AIV in birds’

Response 19: We would like to keep the caption.

Comment 20: Lines 286-287: ‘The infectious agent was anisakidae parasite in all reviewed English language papers (S1 File). ‘ correct: parasites, also other the anisakidae, for example cestods (i.e. Diphyllobothrium)

Response 20: Thank you, we have corrected this.  

Comment 21: Line 456-461: ? very basic

Response 21: We have added more text.

Comment 22: Section 4.3. Risk of bias – it is also rather speculative, without proper support from cited papers. Please, rewrite, add relevant citations to support your opinions.

Response 22: This section has been deleted.

Comment 23: Conclusions: ? very general, not focused on the present study, because some of the topics were not studied/analyzed  here.

Response 23: We have modified the Conclusions section according based on the reviewer’s suggestion.  

Reviewer 3 Report

Dear  Anastasia  and colleagues,
I have found your work timely and worthy of being accepted for publication after clarifiying some details about the boolean operators you have used during your bibliographic research. On the other hand, I have suggested to provide a little bit more detail in your figure captions. Finally, I have suggested to include a quotation about a recent bibliographic review on the impact of ungulates on Artic ecosystems as an example about the need of updating the nowledge in this part of the globe.

I hope that my commenst will be useful for your investigation

Author Response

Dear Reviewer,

Thank you for your valuable comments. Below please find our answers to each comment. In additional, all changes should be visible in the revised manuscript, where we used track change mode to show all changes.

Reviewer 3 – Comments and Suggestions for Authors

Open Review

English language and style

( ) Extensive editing of English language and style required
( ) Moderate English changes required
(x) English language and style are fine/minor spell check required
( ) I don't feel qualified to judge about the English language and style

Dear Anastasia  and colleagues,
I have found your work timely and worthy of being accepted for publication after clarifiying some details about the boolean operators you have used during your bibliographic research. On the other hand, I have suggested to provide a little bit more detail in your figure captions. Finally, I have suggested to include a quotation about a recent bibliographic review on the impact of ungulates on Artic ecosystems as an example about the need of updating the nowledge in this part of the globe.

I hope that my comments will be useful for your investigation

Comment 1: in line with other ongoing research on the interactions between wildlife and the environment in this singular environment (Soininen et al. 2021). Soininen et al. 2021. Location of studies and evidence of effects of herbivory on Artic vegetation: a systematic map. Environmental Evidence. 10:25

Response 1: Reference added.

Comment 2: 2.2. Search strategy and inclusion criteria. It would be interesting here to include your string for research, I mean, the way you have used "AND", "OR" for getting your results.

Response 2: We have included additional supplementary files which list our searches.

Comment 3: Fig. 4. Percentage distribution of mammals analyzed in the evaluated articles (n=231). Pleas, could you be a little bit more specific here? Proportion of studies about the health status of Artic mammal species (%) from XX to XX..

Response 3: This figure was revised and titled: “Fig. 4. Percentage proportion of mammal species in the evaluated articles about the health status of Arctic mammals from 2009 to 2018 (n=231).”. The text under the graph now also contains added percentages and corrections.

Comment 4: Table 2. Top English language journals which published at least three articles on the selected wildlife infections in the circumpolar Arctic. Number of published articles is reported by the host type. Mmmm, do you meant wildlife diseases?. infections included pathogens but many other organisms

Response 4: Thank you a lot for this essential question. In the spirit of One Health, we wish to highlight that not all infections are pathogens to every host, e.g. Trichinella infections in wildlife are mostly asymptomatic and Echinococcus infections in the final host are regarded as apathogenic. Table title change to “Top English language journals which published at least three articles on the selected wildlife infections/diseases in the circumpolar Arctic. Number of published articles is reported by the host type.”

Comment 5: Fig. 6. Main types of infection in mammals… I am not sure whether this information is true. You are showing the most common pathogens studied in wildlife in different countries belonging to the artic, but not the commonest pathogens. Please put units in the Y axis. I am wondering whether it would be worthy to discriminate between birds and mammals here

Response 5: We have now added titles in the vertical axes, and deleted word “main” in the title. We thank the reviewer for the idea stratified figures by host type. However, we prefer to keep birds and mammals together due to the low number of bird articles included in our review.

Round 2

Reviewer 1 Report

Dear authors,

Thank you for your replies. I have some comments related to Future direction section.

Although it is understandable what you wanted to achieve with this section, the overall presentation is poor. 

Please dont just spill out your ideas all over the section - try to reorganize it so the readers can more efficiently get to the point. For example, try to create additional subsections (e.g., 4.2.1 Pathogen mutation and host switch research).